# Second-Generation Antiandrogen Therapy Radiosensitizes Prostate Cancer Regardless of Castration State through Inhibition of DNA Double Strand Break Repair

**DOI:** 10.3390/cancers12092467

**Published:** 2020-08-31

**Authors:** Mohamed E. Elsesy, Su Jung Oh-Hohenhorst, Anastassia Löser, Christoph Oing, Sally Mutiara, Sabrina Köcher, Stefanie Meien, Alexandra Zielinski, Susanne Burdak-Rothkamm, Derya Tilki, Hartwig Huland, Rudolf Schwarz, Cordula Petersen, Carsten Bokemeyer, Kai Rothkamm, Wael Y. Mansour

**Affiliations:** 1Department of Radiotherapy and Radiooncology, University Medical Center Hamburg-Eppendorf, 20246 Hamburg, Germany; m.elsesy@uke.de (M.E.E.); an.loeser@uke.de (A.L.); sally.mutiara@gmail.com (S.M.); s.koecher@uke.de (S.K.); s.meien@uke.de (S.M.); a.zielinski@uke.de (A.Z.); s.burdak-rothkamm@uke.de (S.B.-R.); r.schwarz@uke.de (R.S.); cor.petersen@uke.de (C.P.); k.rothkamm@uke.de (K.R.); 2Department of Tumor Biology, National Cancer Institute, Cairo University, Cairo 11796, Egypt; 3Martini-Klinik Prostate Cancer Center, University Medical Center Hamburg-Eppendorf, 20246 Hamburg, Germany; s.oh-hohenhorst@uke.de (S.J.O.-H.); d.tilki@uke.de (D.T.); h.huland@uke.de (H.H.); 4Institute of Anatomy and Experimental Morphology, University Medical Center Hamburg-Eppendorf, 20246 Hamburg, Germany; 5Department of Oncology, Hematology and Bone Marrow Transplantation with Section of Pneumology, University Medical Center Hamburg-Eppendorf, 20246 Hamburg, Germany; c.oing@uke.de (C.O.); cbokemeyer@uke.de (C.B.); 6Mildred Scheel Cancer Career Center HaTriCS4, University Medical Center Hamburg-Eppendorf, 20246 Hamburg, Germany; 7Department of Urology, University Medical Center Hamburg-Eppendorf, 20246 Hamburg, Germany

**Keywords:** abiraterone acetate, apalutamide, enzalutamide, DNA double strand break repair, prostate cancer, radiosensitization

## Abstract

**Simple Summary:**

The combination of RT and the first generation AR blockers to improve the outcome in prostate cancer remain a matter of controversial debate in clinical trials. In the current study we aim to investigate the effect of three FDA approved second-generation antiandrogens (abiraterone acetate, apalutamide and enzalutamide), as more potent inhibitors of the AR signaling, on the cytotoxicity of RT in pre-clinical models. In vitro and ex vivo analyses revealed a strong radiosensitising effect for the second-generation antiandrogens, regardless of the castration state. The first-generation AR-blocker bicalutamide failed to show any radiosensitising effect. The radiosensitising effect of the second-generation antiandrogens was attributed to the inhibition of DSB repair. Together, we provide a proof-of-principle pre-clinical evidence to rationalize the clinical use of the second-generation antiandrogens to enhance the effect of IR as a potential strategy to improve the outcomes of PCa patients with localized disease who undergo ablative RT.

**Abstract:**

(1) *Background:* The combination of the first-generation antiandrogens and radiotherapy (RT) has been studied extensively in the clinical setting of prostate cancer (PCa). Here, we evaluated the potential radiosensitizing effect of the second-generation antiandrogens abiraterone acetate, apalutamide and enzalutamide. (2) *Methods:* Cell proliferation and agarose-colony forming assay were used to measure the effect on survival. Double strand break repair efficiency was monitored using immunofluorescence staining of γH2AX/53BP1. (3) *Results:* We report retrospectively a minor benefit for PCa patients received first-generation androgen blockers and RT compared to patients treated with RT alone. Combining either of the second-generation antiandrogens and 2Gy suppressed cell growth and increased doubling time significantly more than 2Gy alone, in both hormone-responsive LNCaP and castration-resistant C4-2B cells. These findings were recapitulated in resistant sub-clones to (i) hormone ablation (LNCaP-abl), (ii) abiraterone acetate (LNCaP-abi), (iii) apalutamide (LNCaP-ARN509), (iv) enzalutamide (C4-2B-ENZA), and in castration-resistant 22-RV1 cells. This radiosensitization effect was not observable using the first-generation antiandrogen bicalutamide. Inhibition of DNA DSB repair was found to contribute to the radiosensitization effect of second-generation antiandrogens, as demonstrated by a significant increase in residual γH2AX and 53BP1 foci numbers at 24h post-IR. DSB repair inhibition was further demonstrated in 22 patient-derived tumor slice cultures treated with abiraterone acetate before ex-vivo irradiation with 2Gy. (4) *Conclusion:* Together, these data show that second-generation antiandrogens can enhance radiosensitivity in PCa through DSB repair inhibition, regardless of their hormonal status. Translated into clinical practice, our results may help to find additional strategies to improve the effectiveness of RT in localized PCa, paving the way for a clinical trial.

## 1. Introduction

Prostate cancer (PCa) remains one of the most frequent cancers, and a leading cause of cancer death [1]. Treatment modalities for localized disease include radical prostatectomy, radiation therapy (RT) with or without androgen deprivation therapy (ADT), and active surveillance [1]. Conventional ADT acts by either inhibiting the testosterone production within the testicular stroma through interfering with luteinizing hormone releasing hormone (LHRH) from the pituitary gland or a direct blockade of androgen binding to the androgen receptor (AR). Both approaches block AR signaling, which is the major driver of PCa growth and progression [2]. Currently, various classes of ADT drugs are available, including LHRH agonists and antagonists, and androgen receptor inhibitors (ARIs), such as bicalutamide, flutamide or cyproterone acetate [3,4]. Eventually, ADT prevents the activation and subsequent translocation of the AR to the nucleus, where it acts as a transcription factor, regulating the expression of many target genes that promote prostatic epithelial cell survival and proliferation [4]. ADT has been shown to induce symptom relief and biochemical and objective responses in PCa patients, highlighting the pivotal role of androgens in PCa evolution [5]. Despite the immediate palliative benefits that can be achieved by ADT, the majority of patients relapse within a few years, due to alternative mechanisms of AR signaling, AR amplification or alternative splicing, intratumoral androgen production, or adrenal gland testosterone production. Rising prostate-specific antigen (PSA) values or detectable disease progression despite the appropriate suppression of systemic testosterone levels characterize castration resistance, a major driver of PCa-associated mortality [6,7]. Androgens are still of utmost importance for the growth of castration-resistant PCa (CRPC); that is why CRPC treatment strategies involve novel, second-generation antiandrogenic agents. These differ from LHRH analogues, by blocking specific aspects of extra-gonadal androgen-synthesis and tumoral AR signaling. First-generation antiandrogens established androgen receptor blockade as a therapeutic strategy, but do not completely abrogate androgen receptor activity. Efficacy and potency have been improved by the development of second-generation antiandrogen therapies. These exhibit increased specificity to the AR over other steroidal receptors, act at a higher affinity than the first generation, are exclusively antagonistic to the AR, and in turn, elicit no androgen withdrawal syndrome. 

Several second-generation anti-androgens are currently approved by the Food and Drug Administration (FDA), including the androgen biosynthesis inhibitor abiraterone acetate, which suppresses the CYP17 enzyme, and direct AR blockers, such as enzalutamide, apalutamide and darolutamide, which block AR with 6–9-fold greater affinity than that of the first-generation agent bicalutamide. Notably, abiraterone was found to act as an AR antagonist, which leads to a dose dependent decrease in the AR levels [8,9].

Radiotherapy (RT) is one of the genotoxic modalities that induces various forms of DNA damage. Double strand breaks (DSBs) are considered the most important and toxic lesions induced by ionizing radiation (IR), which, if not repaired or inappropriately repaired, can eventually result in genomic instability and subsequent cell death. A sophisticated DNA-damage response (DDR) machinery, represented by the two main pathways, homologous recombination (HR) and nonhomologous end joining (NHEJ), can ensure fast and appropriate repair of the DSBs. Importantly, tumors with a collapse in the DNA repair capacity of either the HR or NHEJ pathways which can be due to mutations in DDR genes can provide more benefit after radiotherapy [10]. 

RT is an effective local therapy which is used as a curative treatment of localized intermediate or high risk PCa [11]. However, up to 30% of PCa patients show signs of treatment failure within 5 years [12]. The risk of failure of local treatment approaches can be estimated by the D’Amico risk classification stratifying patients as low, intermediate or high risk, based on the known prognostic factors: PSA, Gleason score (GS), and T stage. Moreover, factors associated with intrinsic tumor radioresistance or micro metastatic disease may also contribute to relapses following ablative radiotherapy [13,14]. Possible alternatives to improve RT results include higher radiation doses and agents that optimize the radiation effect [15]. 

Preclinical studies have demonstrated a radiosensitizing role of androgen suppression, arguing towards the combination of RT together with ADT to enhance the therapeutic effect. Despite several clinical studies, the timing and duration of ADT in relation to ablative radiotherapy remain a matter of controversial debate, but ADT has been adopted as a central companion treatment for patients undergoing curative RT for localized disease. For high-risk prostate cancer, long-term ADT for 18 months has been shown to be better than 6 months of ADT in terms of local treatment failure, biochemical relapse rates, distant metastasis-free survival and overall survival (OS) in a large randomized clinical trial [16]. Interestingly, 18 months of ADT were equally effective to 36 months of ADT in the same setting in another large clinical trial [17]. As a consequence, duration (and timing) of ADT in relation to ablative radiotherapy and the mechanisms behind ADT acting as a radiosensitizing treatment need to be further elucidated.

This study aimed to investigate the ability of second generation antiandrogens to enhance the cytotoxic effects and therapeutic ratio of IR, and to determine potential mechanisms underlying this effect. Findings revealed that, regardless of the castration state, second generation antiandrogens, such as abiraterone acetate, apalutamide, and enzalutamide efficiently radiosensitize PCa cells through the inhibition of DNA DSB repair capacity. Our observations provide a mechanistic rationale to study the combination of second generation androgens and locally ablative RT in the clinical setting, to further improve outcomes for patients with localized disease.

## 2. Results

### 2.1. ADT Plus RT Confers a Slight but Not Significant Increase in the Biochemical Relapse-Free Survival of Patients with Intermediate- and High-Risk PCa

Due to the increasing interest for the use of combined antiandrogenic therapy with RT in the management of PCa, we performed a retrospective analysis, employing a cohort of 166 PCa patients treated with RT with or without ADT, between 2008 and 2016 at our institution. The median follow-up was 40 months (range 12–116) and the median age of patients was 73 years old (range 53–80). Further clinicopathological characteristics of the patients are described in Table 1. 

The patients were classified as treated with RT alone (without ADT) or with ADT plus RT (+ADT). Biochemical relapse-free as well as OS were compared in both arms. Neither overall nor biochemical relapse-free survival showed a benefit for the combined treatment compared to the RT alone (Figure 1a,b). 

While the patients with high-risk cancer generally responded worse to RT than those with intermediate cancer as shown in Figure 1c, patients treated with RT and ADT showed a moderate tendency towards increased BCR-free survival in both risk groups. However, we failed to identify any statistical difference between the two groups (±ADT) for either intermediate or high-risk patients (*p* = 0.25). In terms of therapy-related side effects, 59.9% of patients (*n* = 94/155) suffered from low grades (1–2) of acute gastrointestinal toxicity (GIT), while 87.7% (*n* = 136/155) exhibited symptoms of genitourinary toxicity (GUT). There was no significant difference between the two treatment groups regarding the occurrence of GIT (Table 2, *p* = 0.34), and only modest significance regarding GUT (Table 3, *p* = 0.06). 

One common issue between our retrospective study and other reported studies is the use of first-generation antiandrogens, such as bicalutamide or LHRH-analogue. Therefore, a possible explanation for the lack of any significant benefit of the combination strategy could be such that ADT does not completely block the AR-axis.

### 2.2. Second-Generation Antiandrogens Enhance the Response of Prostate Cancer Cells to IR

Thus, we sought to explore the impact of combining second-generation antiandrogens to improve the ionizing radiation effect. To that end, androgen-sensitive LNCaP and castration-resistant C4-2B cells were exposed to IR with a dose of 2 Gy, after 24 h-incubation with two different concentrations of the second-generation antiandrogens; abiraterone acetate (Abi), enzalutamide (ENZA), or apalutamide (ARN509), and the effect on proliferation rate was measured. 

In LNCaP, the second-generation antiandrogens alone exhibited similar growth-inhibiting effects (Figure 2a–c, upper panels), with a clear increase in doubling time (DT) in a dose-dependent manner (Figure 2a–c, lower panels). Irradiating the cells with 2 Gy alone reduced the cell growth to a similar extent as monotherapy with the respective antiandrogen, as illustrated by similar DTs. Interestingly, combining either of the indicated novel antiandrogens and 2 Gy further suppressed the growth rate of LNCaP, cells as exemplified by increased DTs of at least 1.5-fold for combination, with the lower concentration of antiandrogen and up to 2.3-fold with the therapeutic concentration, indicating a potential cooperative effect between these novel antiandrogens and IR. 

Strikingly, in castration-resistant C4-2B cells, we observed similar proliferation inhibitory effects (Figure 2d–f, upper panels) and increased DTs (Figure 2d–f, lower panels) as in androgen-sensitive LNCaP cells (Figure 2a–c). In order to consolidate these findings, castration-resistant 22-RV1 cell lines were employed, and the colony formation assay was used to assess the effect of the afore-mentioned combination therapy strategies on clonogenic cell survival. As illustrated in Figure 3a, the combination of antiandrogen with 2 Gy strongly inhibited the survival of 22R-V1 cells, despite the moderate effect of single therapy with either antiandrogens or 2 Gy alone (Figure 3a). In keeping with the radiosensitization idea, we reported a significant radiosensitizing effect on 22R-V1 cells, upon combining 5 µM abiraterone acetate and different IR doses (Figure 3b). Importantly, this strong radio-sensitization effect was specific to the second-generation antiandrogens, while the first-generation of antiandrogen, bicalutamide (10 µM), failed to further enhance the cytotoxic effects of IR compared to either bicalutamide or 2 Gy alone in 22R-V1 (Figure 3b), as well as in both LNCaP and C4-2B cells (Figure 3c,d). Notably, no difference was reported between the extents of radiosensitization mediated by the tested second-generation antiandrogens (Appendix A). Together, these data suggest that second-generation antiandrogens can more efficiently radiosensitize PCa cells, regardless of castration state.

### 2.3. Second Generation Antiandrogens Escalate the IR Effect About 2 Times 

The use of RT doses higher than the conventional IR doses of 70 to 72 Gy could increase the ability to sterilize and thereby cure PCa. However, additional RT doses would increase normal tissue toxicity, which limits this escalation strategy. Therefore, it is crucial to develop a strategy to intensify the effect of the IR without escalating the dose. Based on the previous investigation that ADT reduces the dose of RT required to control 50% of Shinonogi adenocarcinoma tumors [18], we compared the effect of escalating IR doses and the combination therapy with the second generation of antiandrogens. Therefore, we escalated the IR dose to 5 and 10 Gy, and compared the effect of these escalated radiation doses with the effect of combining 2 Gy with different concentrations of novel antiandrogens. As expected, a strong IR dose-dependent growth inhibition effect was found in both LNCaP and C4-2B cell lines (Appendix A). In both cell lines, the growth-inhibitory effect of combining 2 Gy and therapeutic concentrations of the utilized antiandrogens was similar to that of 5 Gy alone, as evidenced by (i) no, or at least, very little difference between the average of the inhibitory effects of both treatment settings, after 3 days, 6 days or 10 days (Figure 4a,b), and (ii) the similar increase in DTs (3.05 d, 2.8d, 2.9d and 2.7d for 5Gy, Abi+2 Gy, ARN+2 Gy and ENZA+2 Gy, respectively) (Figure 4c). In order to further verify this, LNCaP and C4-2B cells were treated with 5 µM abiraterone acetate and IR (0, 1, 2, 5, 10 Gy), either individually or combined, and effects on survival were analyzed using agarose CFA. Again, a radiosensitizing effect was reported upon pre-treatment with abiraterone acetate in both cell lines, and interestingly the effect of abiraterone + 2 Gy was similar to that of 5 Gy alone (Appendix A). Again, bicalutamide failed to radiosensitize either LNCaP or C4-2B cells (Appendix A). These data suppose a beneficial clinical outcome from the use of second-generation antiandrogens, either to (i) reduce IR doses, hence alleviating the adverse effects from higher IR doses, or (ii) to escalate the effect of the same RT dose.

### 2.4. Appropriate Choice of Antiandrogen to Maximize the Therapeutic Effect of IR in Acquired AHT-Resistant PCa Cells

The above data reveal that second-generation anti-androgens can efficiently radiosensitize even the CRPC cells. Since the CRPC is a very heterogeneous disease, and in order to more generalize our findings, we sought to employ PCa cell lines that had acquired resistance to therapeutic agents that target the AR axis [19,20], including (i) three subclones from the LNCaP cells which mimic hormone ablation-resistance (LNCaP-abl), are resistant to abiraterone acetate (LNCaP-Abi), or to apalutamide (LNCap-ARN509) and (ii) one C4-2B subclone which is resistant to enzalutamide (C4-2B-ENZA). 

Firstly, the resistant phenotypes of these cells were confirmed. In parental LNCaP cells, the cell growth rates were significantly inhibited with an approximately 2-fold increase in the DT in the presence of therapeutic concentrations of either of the antiandrogens. The resistant LNCaP sub-clones however, showed no effect on cell growth (Figure 5a–c, upper panels) and no difference in the DTs (Figure 5a–c, lower panels) when cultured in hormone-ablated medium (from 2.0 to 2.2 days), or treated with either 10 µM abiraterone acetate (from 3.836 to 4.132 days, Figure 5b) or 20 µM ARN509 (from 2.6 to 2.8 days, Figure 5c) for 10 days. Likewise, C4-2B-ENZA did not exhibit any significant change in the growth profile or DT (from 2.4 to 2.6 days) after treatment with 20 µM enzalutamide, while the proliferation of parental C4-2B cells was dramatically suppressed with a 1.7-fold increase in the DT (Figure 5d).

Next, we investigated the effect of the different antiandrogens on the radiosensitivity of the resistant sublines. To that end, cells were treated with the indicated antiandrogens and 2 Gy either individually or combined, and the effect on cell growth was monitored by cell counting at 3, 6 and 10 days post treatment. As indicated in Figure 6a, in addition to the expected resistance to hormone ablation, LNCaP-abl cells were resistant to the other novel antiandrogens with no or very minor effect on DTs. However, combining 2 Gy with ARN509 or enzalutamide, but not abiraterone, significantly inhibited cell growth compared to the single use of antiandrogens or 2 Gy alone (Figure 5a). Enhanced radiosensitivity was also reported in LNCaP-abl cells grown in hormone-ablated medium with a 1.6-fold increase in DT compared to the same cells grown in hormone proficient medium (Figure 4a,e), indicating that ADT may radiosensitize even ADT-resistant cells. This was further confirmed in the other LNCaP-derived resistant subclones. 

LNCaP-abi cells exhibited, as expected, no effect on cell growth upon treatment with 10 µM abiraterone, while enzalutamide and ARN509 slightly decreased the cell growth of these cells, as evidenced by increased DTs (from 3.173 days to 4.638 days and 3.884 days for ARN and ENZA, respectively). Strikingly, pre-treatment with any of the second generation antiandrogens, also including abiraterone, enhanced the cytotoxic effects of 2 Gy, as illustrated by compromised cell proliferation (Figure 6b) and a dramatic increase of DTs after combined treatments (2.9-fold, 5-fold and 7.2-fold after combination with abiraterone-acetate, apalutamide or enzalutamide, respectively) (Figure 6e). LNCaP-ARN509 was resistant to both novel ARI ARN509 and enzalutamide, but could be radiosensitized upon treatment with the same drugs as well as abiraterone (Figure 6c,e). 

While both ARN509 and enzalutamide showed no inhibitory effect on the proliferation of C4-2B-ENZA cells, abiraterone acetate treatment conferred a growth suppression similar to that caused by 2 Gy. Interestingly, these cells demonstrated an ultimate growth suppression when 10 μM abiraterone acetate was combined with 2 Gy, with about 2-fold increase in DT (Figure 6d,e). Notably, bicalutamide showed contradictory effects on the radiosensitivity of the hormone resistant sub-clones. Bicalutamide treatment showed radioprotective activities on LNCaP-abi and LNCaP-ARN cells, by increasing the proliferation rate of these cells, as evidenced by decreasing the DTs of LNCaP-abi and LNCaP-ARN cells from 3.1 to 1.9 days and from 3.6 to 2.1 days, respectively (Appendix A). This radio-protective effect can be explained by a previously described agonistic effect of bicalutamide on AR signaling [19], which might lead to the stimulation of the repair capacity. On the other hand, bicalutamide alone showed a stronger growth inhibitory effect on C4-2B-ENZA cells compared to IR alone. Combining bicalutamide and IR resulted in a slight increase in DT of C4-2B-ENZA cells from 2.3 to 2.6 days (Appendix A). Collectively, these data further confirm that second generation antiandrogens can be used to efficiently potentiate the cytotoxic effects of IR, even in CRPC.

### 2.5. Second-Generation Antiandrogens Radiosensitize PCa Cells Through the Inhibition of DSB Repair

DNA DSBs are considered the most lethal type of DNA damage induced by IR, and the DSB repair capacity of cells can determine their radiosensitivity, as well as the radiosensitizing effect of a specific treatment. To address whether the inhibition of DSB repair is the mechanism underlying the antiandrogen-mediated radiosensitization, DSBs were monitored using γH2AX and 53BP1 in androgen-sensitive LNCaP and castration-resistant C4-2B cells after treatment with the individual anti-androgens and 2 Gy, either alone or combined (Figure 7). 

Antiandrogens alone did not increase the number of γH2AX or 53BP1 foci compared to untreated controls. The exposure to 2 Gy increased the number of γH2AX and 53BP1 at 1 h (28.1 ± 2.1 foci/cell), and no further induction was observed 1 h post 2 Gy in addition to AHT. Remarkably, pretreatment of LNCaP cells with 5 µM Abi, 10 µM enzalutamide or 20 µM ARN509 resulted in a significant increase in the number of individual (Appendix A) and colocalized γH2AX and 53BP1 foci (Figure 7a,c) at 24 h post 2 Gy (*p* < 0.0001), indicating a severe inhibition of DSB repair. Hence, LNCaP cells exposed to the novel antiandrogens were compromised in their ability to repair the IR-induced DSBs. These data were confirmed in C4-2B cells (Appendix A and Figure 7b,d), showing again that antiandrogens exhibited no difference in the induction of DSBs, i.e. at 1 h post 2 Gy (*p* = 0.2), but significantly enhanced the number of residual γH2AY/53BP1 foci at 24 h post 2 Gy (*p* < 0.0001). In agreement with the absence of bicalutamide induced radiosensitization, pretreating either LNCaP or C4-2B cells with 10µM bicalutamide did not increase the number of colocalized γH2AX and 53BP1 foci (Figure 7) at 24 h post 2 Gy (*p* = 0.74 and *p* = 0.80, for LNCaP and C4-2B cells, respectively). Next, we sought to identify the repair pathway which is inhibited upon abiraterone treatment. In order to address this issue, LNCaP and C4-2B cells were treated with 5 µM abiraterone acetate for 24 h, and then with either 5 µM of either NU55933 (ATM inhibitor, ATMi) or NU7026 (DNAPK inhibitor, DNAPKi) for 2 h, before being irradiated with 2Gy, and subsequently, γH2AX and 53BP1 foci were monitored at 1 h and 24 h post-IR. As illustrated in Appendix A, combining ATMi and abiraterone did not further increase the number of residual γH2AX/53BP1 compared to either ATMi or abiraterone. On the other hand, DNAPK inhibition increased the number of residual γH2AX/53BP1 significantly more than abiraterone treatment alone. Furthermore, combining DNAPKi and abiraterone show a tendency of a synergistic increase in the number of residual unrepaired DSBs (Appendix A). Together, these data indicate that abiraterone probably inhibits HR, but not NHEJ.

It is established that AR activity regulates the cell cycle and its inhibition causes a permanent G1 arrest [21]. Therefore, it was critical to assess whether the observed radiosensitization is cell cycle-dependent. To that end, the cell cycle profile was assessed in LNCaP (upper panel) and C4-2B (lower panel) after different time intervals post-combined treatment, and was compared to the effect of either IR or antiandrogens alone. As illustrated in Appendix A, irradiation arrested LNCaP cells in the S/G2 phase, which was resolved at later time points. Combining either of the novel antiandrogens with IR did not change the cell cycle profile compared to IR alone. Similar results were also reported in C4-2B cells. Furthermore, the analysis of apoptosis revealed no difference in apoptosis rates amongst the different treatment conditions (Appendix A). Together, this reveals that, regardless of castration state, the novel antiandrogens radiosensitize PCa cells through inhibition of DSB repair and independently of cell cycle redistribution or the induction of apoptosis. 

### 2.6. Abiraterone But Not Bicalutamide Reduces Repair Capacity of IR-Induced DSBs in Fresh Prostate Cancer Tissues 

In order to further confirm the novel antiandrogen-mediated inhibition of DSB repair and hence the radiosensitization effect, we employed the previously described functional ex vivo assay, which enables monitoring DSB repair in fresh tumor tissues [22]. We obtained 26 fresh punch biopsies from 17 high-risk PCa patients undergoing radical prostatectomy at the Martini-Klinik Hamburg, Germany. Specimens were routinely examined by a pathologist, and only tissue samples with confirmed malignancy were included in this study. Cell viability and oxygenation were confirmed by monitoring EdU^+^ cells and pimonidazole staining as described [22]. Two patient samples were excluded from the analysis because they did not follow the previously described basic selection criteria [22]. The remaining punch biopsies were treated with 10 µM abiraterone for 24 h before irradiation with 2 Gy. γH2AX and 53BP1 foci were analyzed 1 h and 24 h later (Figure 8a).

As illustrated in Appendix A, the slices demonstrated no difference in the number of γH2AX and 53BP1 upon treatment with abiraterone alone; however, this number increased upon irradiation with 2 Gy. Most tumor slices displayed significantly elevated numbers of residual γH2AX and 53BP1 foci after 24 h of combined treatment with Abi and 2 Gy, compared to the single treatment. Analysis of the abiraterone acetate-induced radiosensitization enhancement ratio (AbiER) revealed that tumor samples from 13 out of 15 PCa patients (67%) showed at least a 2-fold increase in their AbiER index, with minor alterations between both γH2AX (Figure 8a) and 53BP1 (Figure 8b) markers. Notably, two punch biopsies obtained from the same patient exhibited consistent results. Bicalutamide failed again to enhance the radiosensitivity in freshly collected prostate tumor tissues from 8 PCa patients (BicaER) (Figure 8d and Appendix A). The mean standard error of all samples was set as a threshold for each DSB marker. Together, this further confirms the finding that the novel antiandrogens, such as abiraterone acetate, radiosensitize PCa cells by suppressing DSB repair capacity. 

## 3. Discussion

The combination of ADT and RT has been studied extensively in the clinical setting of localized PCa. Moreover, there is a strong rationale from preclinical models, highlighting a radiosensitizing effect of AR signaling inhibition. Clinically, the beneficial effect for an ADT plus RT combination depends on several factors, including the histopathological features of the tumor, tumor stage, PSA-level, and duration of ADT, among others. For low risk PCa patients treated with ADT plus RT, the RTOG 94-08 trial failed to show any significant improvement in disease control. There was, however, a significant decline in the biochemical failure rate, favoring the combined treatment approach [23]. Although a similar effect was observed in several other studies, the impact on OS for PCa patients receiving RT with ADT as compared with those receiving RT alone is less evident [24,25], except for high risk patients [16]. EORTC 22863 also showed a survival benefit for locally advanced PCa following combined treatment [26]; however, other studies could not recapitulate this benefit for locally advanced PCa patients [27,28]. RTOG 92-02 revealed no significant improvement in 10-year survival for PCa patients, except for those with a Gleason score of 8 or higher [28]. However, the Quebec L200 trial failed to recapitulate the results of these studies concerning biochemical failure rates [29]. Consequently, trial results are heterogeneous, which is at least, in part, ascribable to heterogeneous patient populations, end points and modalities of treatment applied, but a beneficial effect of ADT on RT activity seems highly likely. Therefore, concurrent ADT plus locally ablative RT has been adopted in clinical guidelines and routine care pathways, at least for intermediate and high risk localized PCa patients.

In the present study, a retrospective analysis of patients undergoing definitive radiotherapy for localized PCa at our institution confirmed a modest but insignificant improvement in biochemical failure-free survival of PCa receiving combined ADT and RT over RT alone. Assuming a correlation of impaired androgen signaling with enhanced radiosensitivity, and in view of the controversial clinical results, we opined that more effective AR signaling inhibition by second-generation antiandrogens would enhance the cytotoxicity of RT in pre-clinical models.

In the current study, we used the second generation antiandrogens abiraterone acetate, apalutamide and enzalutamide, all of which are FDA approved for clinical use in metastatic hormone-sensitive and castration-resistant prostate cancer (apalutamide in the latter setting only). The concentrations used for pre-clinical in vitro assessment are also in the range of clinically achievable steady state plasma concentrations (Cmax) [30]. Indeed, the concentrations of the drugs intra-tumoral are basically much lower than the Cmax. However, this study provides the evidence base for the clinical use of the aforementioned drugs as radiosensitizers in prostate cancer.

Taken together, our results revealed a stronger growth inhibition induced by combining one of the second generation antiandrogens: abiraterone acetate, enzalutamide or apalutamide, along with 2 Gy compared to antiandrogen or IR, alone or IR in combination with bicalutamide as a first-generation AR blocker. In fact, bicalutamide did not enhance RT-induced growth delay in our cell line models. In contrast, second generation antiandrogens profoundly intensified the effect of IR, as illustrated by a consistent increase in the DTs of irradiated PCa cells that had been treated with second-generation antiandrogens. We provide here proof-of-principle pre-clinical in vitro and ex vivo evidence to rationalize the clinical use of the second-generation antiandrogens to enhance the effect of IR as a potential strategy to improve the outcomes of PCa patients with localized disease who undergo ablative RT.

The inhibition of DNA DSB repair capacity was found to underlie the mediated radiosensitization effect of second-generation antiandrogens. This was evidenced by a significant increase in the number of both γH2AX and 53BP1 foci remaining after 24 h post 2 Gy upon pretreating PCa cells, with either of the second-generation antiandrogens. Importantly, repair inhibition was further demonstrated in 22 freshly collected tumor tissue samples from 15 PCa patients treated with abiraterone acetate, before being ex vivo irradiated with 2 Gy. Previous studies reported a tight connection between DNA damage repair and AR signaling through hormone-mediated regulation of several DNA repair genes [31,32,33,34].

Another key conclusion in the current study is that the ADT-mediated radiosensitization was found to be independent of the hormone sensitivity state of the PCa cells. This was derived from the findings that (i) the castration-resistant C4-2B cells showed a similar radiosensitization effect upon treatment with any of the second-generation antiandrogens, compared to the hormone-sensitive prostate cancer (HSPC) LNCaP cells; (ii) the DT was increased similarly in both C4-2B and LNCaP cells, and furthermore, (iii) different specific ADT-resistant clones established from the HSPC cell line LNCaP were sensitized to IR by at least one of the second generation antiandrogens, interestingly including the ADT to which they are resistant. Notably, Goodwin et al. [35], reported—in contrast to our data—a smaller effect on IR sensitivity for the second-generation of antiandrogen, enzalutamide in the CRPC C4-2 cells. This discrepancy may be explained by the fact that they used a lower enzalutamide concentration to radiosensitize C4-2 cells. Another possible explanation could be that, given its dual role in inhibiting AR signaling [8,9], abiraterone might be a more potent radiosensitizing agent compared to enzalutamide and apalutamide. In keeping with this possibility, we showed here that abiraterone radiosensitized the hormone resistant cells more efficiently than enzalutamide and apalutamide.

IR is a local therapy with side effects restricted to tissues lacking AR and several second-generation ADT agents, such as abiraterone, apalutamide, enzalutamide and darolutamide; these are approved by the FDA for localized CRPC (M0 CRPC) [36], and this would rationalize the use of such drugs in combination with RT to control the localized HSPC, and probably prevent the progression to CRPC, with minimal side effects. 

The STAMPEDE trial includes groups of PCa patients receiving a combination of either the first-generation ADT with RT, docetaxel and abiraterone acetate, or the first-generation ADT with enzalutamide and abiraterone acetate (NCT00268476). Although the data published so far demonstrated a benefit for the first combination arm, namely; abiraterone acetate + standard care treatment including RT over Abiraterone acetate + the standard care without RT, this data recommended the use of RT, but does not provide any clue for the radiosensitization effect of abiraterone. However, it provides the rationale for the use of abiraterone acetate in combination with RT as a treatment option. Additionally, the COUAA-31 trial (NCT00638690) reported that abiraterone acetate plus radiation was safely co-administered to patients with a perceived advantage in the palliative bone metastasis response. Abiraterone acetate with RT has been further tested in two separate phase 3 clinical trials, including ERA-223 (NCT02043678), and in the ongoing PEACE 1 trial as well [37]. Indeed, further randomized clinical trials are required to assess the effect of combining the second generation antiandrogens and RT for PCa patients. 

The current study provides the rationale for this treatment regime for PCa patients, through combining second generation antiandrogens and RT for low risk/locally advanced PCa patients, which might prevent progression to CRPC.

## 4. Materials and Methods 

### 4.1. Patients and Retrospective Study Design 

A total of 166 patients (median age: 73 years, range 53–80) with localized prostate cancer underwent high dose-rate brachytherapy (HDR-BT), combined with subsequent external beam radiotherapy between 2008 and March 2016, at the Department of Radiotherapy and Radiooncology of the University Medical Center Hamburg-Eppendorf. Only patients who signed written informed consent and those with complete data sets were included in this analysis. HDR-BT was delivered with 9 Gy/fraction on days 1 and 8 with an iridum-192 source, while EBRT was administrated with 1.8 Gy/fraction to a target dose of 50.4 Gy. Among these patients, 46 (27.7%) received neoadjuvant and/or concomitant ADT. Side effects were classified according to the toxicity criteria of the Radiation Therapy Oncology Group (RTOG) and the European Organization for Research and Treatment of Cancer (EORTC), as previously described [38].

### 4.2. Cell Culture, Drugs, and X-Irradiation

LNCaP and C4-2B prostate cancer cells (ATCC, Manassas, VA, USA) were grown in DMEM (Gibco, Invitrogen, Karlsruhe, Germany), supplemented with 10% fetal calf serum, 100 U/mL penicillin and 100 mg/mL streptomycin at 37 °C, with 10% CO2. LNCaP-ARN509, LNCaP-abi, C4-2B-Enza cells (kindly pro-vided by Prof. CP Evans, UC Davis School of Medicine, Sacramento, CA, USA) were maintained in media containing the corresponding drug to which they are resistant as previously described [17,18]. Abiraterone acetate and apalutamide were kindly provided by Janssen Cilag GmbH, Neuss, Germany. Bicalutamide and enzalutamide were purchased from Selleckchem, Germany. LNCaP-abl cells (a gift from Prof. Culig, Medical University Innsbruck, Austria) were grown in DMEM, supplemented with 10% Charcoal Stripped FBS (Sigma-Aldrich, Deisenhofen, Germany). All LNCaP-abl cells experiments were performed in poly-L-lysine coated 6-well culture plates. All cell lines tested negative for mycoplasma contamination. Irradiation was performed as previously described (200 kV, 15 mA, additional 0.5mm Cu filter at a dose rate of 0.8 Gy/min) [19]. To inhibit the kinase activity of ATM and DNAPK, 5µM KU55933 (Selleckchem, Germany) and 5 µM NU7026 (Selleckchem, Germany) were used, respectively.

### 4.3. Proliferation Assay

Cells were plated in triplicate in 6-well plates, cultured in the appropriate growth media and allowed to attach overnight before treatment with the indicated drugs. For hormone ablation experiments with LNCaP-abl subclone, cells were seeded in FBS full medium for 18–24 h before changing to steroid-deprived medium (CS-FCS) for the indicated time points, with or without irradiation. To assess the effect of any treatment regimes, the cell number was determined via Beckman Coulter cell counter (Life Science, Krefeld, Germany) at 3, 6, and 10 days post-treatment. In all experiments, media with or without drugs were changed twice in the 10-day treatment course.

### 4.4. Colony Formation Assay

Cellular survival was determined via colony formation assay, as previously described [39,40]. Briefly, cells were plated at 200 cells per well in a 6-well plate, in the presence of 5 µM abiraterone acetate or 10 µM bicalutamide. After 24 h, cells were X-irradiated (RS225 research system, GLUMAY MEDICAL, UK at 200kV, 15 mA) and maintained for 2–3 weeks. Colonies were thereafter fixed in 70% ethanol and stained in 0.1% crystal violet. Cellular survival was defined as the ability to form colonies containing at least 50 cells. For agarose CFA, cells were mixed in 0.3% agarose in DMEM with 10% FCS and plated at 10000 cells/well, onto 6-well plates containing a solidified bottom layer (0.6% agarose the same growth medium). After 14 days, colonies were stained with 0.5 mg/mL MTT (Sigma-Aldrich), and photo-graphed using REBEL Microscopy (ECHO, San-Diego, CA, USA). Colonies were then counted using Image-J. Surviving fractions (SF) were calculated by normalization to the plating efficiency of the un-irradiated control. DMSO was used as a control at the same concentration. 

### 4.5. Immunofluorescence

Treated cells on coverslips were washed once with cold PBS and fixed with 4% para-formaldehyde/PBS for 10 min. Fixed cells were permeabilized with 0.2% Triton X-100/PBS on ice for 5 min and incubated for 1 hour at room temperature with primary antibodies: Mouse monoclonal anti-phospho-S139-H2AX antibody (Millipore, Berlin, Germany), at a dilution of 1:500 and rabbit polyclonal anti- 53BP1 antibody (Novus), at a dilution of 1:500. After being washed three times with cold PBS, the cells were incubated for 1h with secondary anti-mouse Alexa-fluor594 (Invitrogen), at a dilution of 1:500 or anti-rabbit Alexa-fluor488 (Invitrogen) at a dilution of 1:600. The nuclei were counterstained with 4′-6-diamidino-2-phenylindole (DAPI, 10 ng/mL). Slides were mounted in Vectashield mounting medium (Vector Laboratories). 

Immunofluorescence of cultured tumor tissue was performed as previously described [22]. Fluorescence microscopy was performed using the Zeiss AxioObserver.Z1 microscope (objectives: x20, resolution 0.44 µm; Plan Apo 63/1.4 Oil DICII, resolution 0.24 µm; and filters: Zeiss 43, Zeiss 38, Zeiss 49). Z-stacks of semi-confocal images were obtained using the Zeiss Apotome, Zeiss AxioCam MRm and Zeiss AxioVision Software. For DSB analysis, fields of view were taken per time point or treatment with a minimum of 100 cells (xenograft) or 50 cells (primary tumor). All stainings were performed in duplicates. DSBs were analyzed using ImageJ and DAPI-based image masks, and normalized to single nucleus values [22].

### 4.6. Cell Cycle Analysis

For cell cycle analysis, treated cells were harvested and fixed with 80% cold ethanol (−20 °C). After washing, the DNA was stained with propidium iodide solution containing RNase A. Cell cycle distribution was monitored by flow cytometry (FACS CANTO 2, BD Bioscience Systems, Heidelberg, Germany) and analyzed using Mod-Fit software (Verity Software House). 

### 4.7. Apoptosis Quantification

Apoptosis was investigated by detection of caspase activity utilizing the FAMFLICA ™ Poly Caspases Assay Kit (Immunochemistry Technologies, Bloomington, MN, USA), according to the manufacturer’s instructions. Flow cytometric analysis was performed on a FACS Canto with FACS Diva Software (Becton Dickin-son, Toronto, ON, Canada). Staurosporine (Sigma S6942) was used as a positive control with a final concentration of 1 µM for at least 12 h incubation.

### 4.8. Patient Sample Collection

Fresh PCa tissue was obtained from patients with high-risk PCa according to D’Amico risk stratification undergoing radical prostatectomy at Martini-Klinik, Prostate Cancer Center Hamburg, Germany. After resection, 1–2 punch biopsies were taken by the surgeon in palpable tumor areas. The biopsies were collected in culture media and immediately taken to the laboratory. Anonymized biopsies were processed within 30 minutes after resection. The laboratory received a final pathology report containing the Gleason score, PSA status and age of each anonymized patient for clinical analysis. 

### 4.9. Tissue Slice Cultures

Tissue slice cultures were prepared as described [22]. Briefly, 300 µm slices were cut using the MacIllvine tissue chopper and placed on Millicell® cell culture inserts (0.4 µm, 30 mm diameter, Merck, Soden, Germany), which were inserted in 6 well dishes containing 1 mL Dulbecco’s modified Eagle medium (DMEM; Gibco-Invitrogen), supplemented with 10% fetal calf serum (FCS) and incubated at 37 °C. Prior to ex vivo treatment, the tissues slices were incubated for one day for recovery and re-oxygenation. To monitor proliferation, un-irradiated slice cultures were incubated with 5-ethynyl-20-deoxyuridine (EdU, 1:1000; Click-iT Assay Kit, Invitrogen) overnight for 16 h. All slices were additionally treated with pimonidazole (200 µM, Hypoxyprobe), 2 h before fixation, to monitor hypoxia. To analyse the effect of abiraterone or bicalutamide on IR, slices were treated with 5µM abiraterone or 10µM bicalutamide for 24 h before irradiation, using a Gulmay X-ray source (200 kV, 15 mA, additional 0.5mm Cu filter, dose rate of 0.8 Gy/min). 

### 4.10. Graphs and Statistics

Unless stated otherwise, experiments were independently repeated at least three times. Data points represent the mean ±SEM of all individual experiments. Survival curves were deduced by means of the Kaplan–Meier method and comparisons were made by log-rank test. To estimate a hazard ratio (HR) of an occurring event, Cox proportional hazards regression model was applied (at a 95% confidence interval (CI)). A *p*-value of <0.05 was regarded statistically significant. Statistical analyses, data fitting and graphics were performed with the GraphPad Prism 7.0 pro-gram (GraphPad Software), SPSS Statistics 25 software (IBM Inc. SPSS Statistics, Armonk, NY, USA) and MedCalc 18.11 (MedCalc Software, Ostend, Belgium). 

### 4.11. Ethical Approval

This study was in accordance with the World Medical Association Declaration of Helsinki, and the guidelines for experimentation with humans by the Chambers of Physicians of the State of Ham-burg (“Hamburger Ärztekammer”). All patients gave informed consent for their excised prostate specimens to be used for research purposes. All experiments were approved (Approval No. PV 7007) by the Ethics Committee of the Chambers of Physicians of the State of Hamburg (“Ham-burger Ärztekammer”). 

## 5. Conclusions

We present a mechanistic rationale for the use of second-generation antiandrogens to radio-sensitize prostate tumors via the inhibition of DSB repair, interestingly regardless of castration state. The potential for the novel antiandrogens as standalone therapeutic agents seems to have plateaued for use in advanced PCa. It is far more likely that the next wave of therapeutic investigation will be focused on the combination of antiandrogen therapy, with other treatments such as radiotherapy and chemotherapy. The current study provides the proof-of-principle for the currently ongoing clinical trials, and paves the way to initiate additional ones.

## Figures and Tables

**Figure 1 cancers-12-02467-f001:**
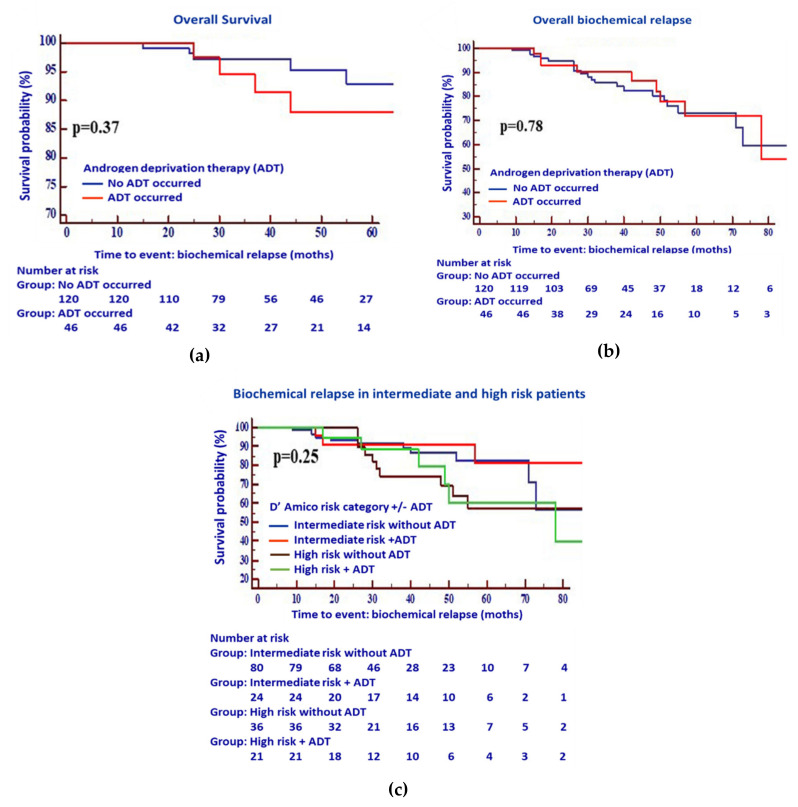
Kaplan–Meier estimates for (**a**) overall survival following RT with ADT (*n* = 65) or without ADT (*n* = 138), (**b**) the overall biochemical relapse of intermediate risk vs high risk PCa patients, and (**c**) the biochemical relapse in intermediate and high risk PCa patients following RT, with or without ADT. A multivariable Cox proportional hazards model did not show any statistically significant differences between the biochemical relapse of patients treated with ADT plus RT and those treated with RT alone.

**Figure 2 cancers-12-02467-f002:**
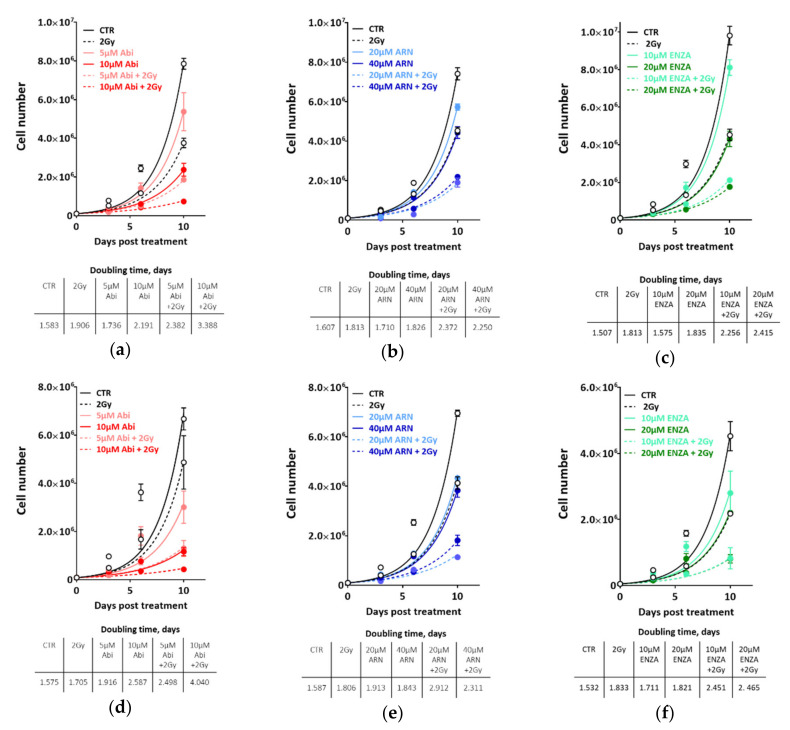
ADT potentiates the cytotoxicity of ionizing radiation in both hormone sensitive LNCaP and hormone resistant C4-2B cells. Cell number was determined in LNCaP (**a**–**c**) or C4-2B cells (**d**–**f**) on days 0, 3, 6 and 10 post treatment, with the indicated concentrations of abiraterone acetate (**a**,**d**), apalutamide (ARN) (**b**,**e**) or enzalutamide (ENZA) (**d**,**f**). Cell doubling time in days was calculated for each treatment, by fitting exponential growth curves using GraphPad Prism 7. Shown are means ±SEM of at least three independent experiments.

**Figure 3 cancers-12-02467-f003:**
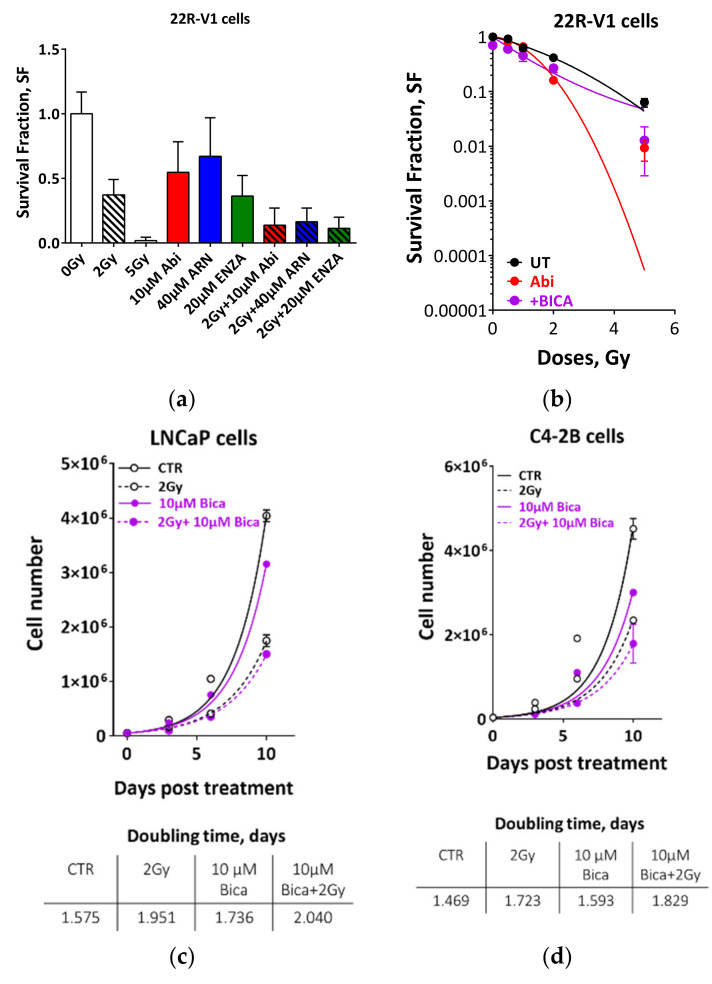
Second generation antiandrogen therapy but not bicalutamide potentiates the cytotoxicity of ionizing radiation in PCa cells. (**a**) 22R-V1 were treated with the indicated concentrations of second-generation antiandrognes and 2Gy either alone or combined and cell survivals were measured using colony forming assay. (**b**) 22R-V1 cells were treated with either 5µM abiraterone acetate or 10µM bicalutamide for 24 h before irradiated with the indicated doses, and the survival fractions (SFs) were measured using colony forming assay. (**c**,**d**) Upper panels: Cell number was determined in LNCaP (**c**) and C4-2B (**d**) cells on days 0, 3, 6 and 10 post treatment with 10 µM bicalutamide and 2 Gy either individually or combined. Lower panels: Cell doubling time in days was calculated for each treatment by fitting exponential growth curves using GraphPad Prism 7. Shown are means ±SEM of at least three independent experiments.

**Figure 4 cancers-12-02467-f004:**
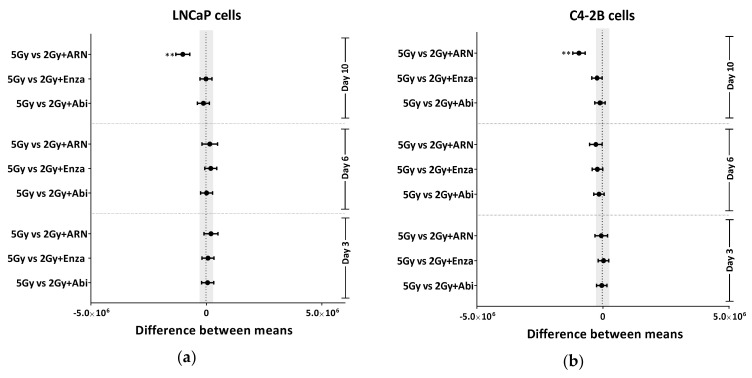
Second generation antiandrogen therapy enhances the IR effect at least 2-fold. Tukey’s multiple comparisons test was used to compare between the effect of 5 Gy and the different indicated treatments on the growth inhibition in (**a**) LNCaP and (**b**) C4-2B cells. Significance was measured using two-way ANOVA test. (**c**) Cell-doubling time in days was calculated for the indicated treatments by fitting exponential growth curves using GraphPad Prism 7. Shown are means ±SEM of at least three independent experiments. Significance is indicated as ** for *p* < 0.001.

**Figure 5 cancers-12-02467-f005:**
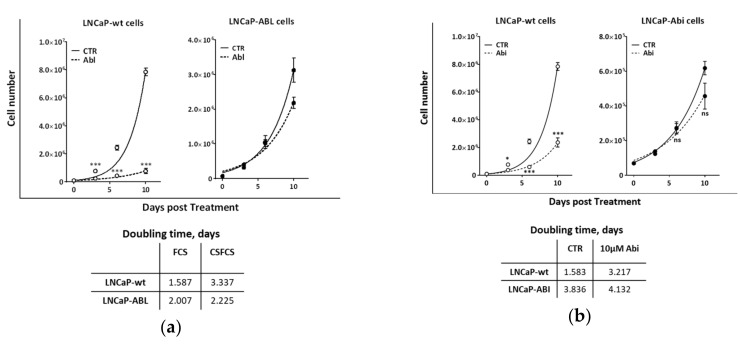
Resistance phenotype of ADT resistant PCa cells. Cell number was determined on days 0, 3, 6 and 10 post treatment with (**a**) hormone ablation, (**b**) 10 µM abiraterone acetate, (**c**) 40 µM apalutamide, or (**d**) 20 µM enzalutamide. Cells used are LNCaP (LNCaP-wt) cells, and their resistant sublines to: hormone ablation (LNCaP-abl), abiraterone acetate (LNCaP-abi), or apalutamide (LNCaP-ARN509), as well as wildtype C4-2B (C4-2B-wt) cells, or their resistant subclone to enzalutamide (C4-2B-ENZA). Cell-doubling time was calculated for each treatment by fitting exponential growth curves using GraphPad Prism 7. Shown are means ±SEM of at least three independent experiments. Significance is indicated as * for the *p* < 0.05, 0.001 and *** for *p* < 0.0001, ns: not significant.

**Figure 6 cancers-12-02467-f006:**
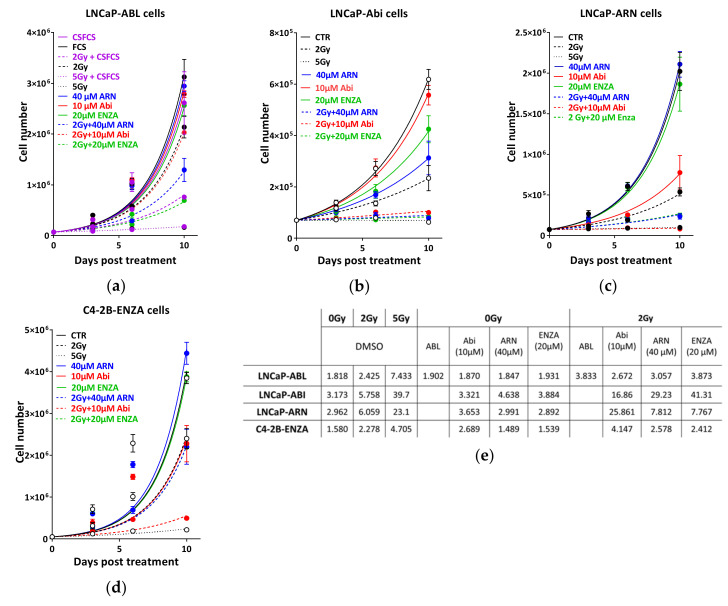
Second generation ADT potentiates the cytotoxicity of ionizing radiation in castration resistant PCa cells. Cell number was determined on days 0, 3, 6 and 10 post treatment with the indicated treatments in LNCaP resistant sublines, to either (**a**) hormone ablation (LNCaP-abl), (**b**) abiraterone acetate (LNCaP-abi), or (**c**) apalutamide (LNCaP-ARN509), as well as (**d**) enzalutamide resistant C4-2B subclone (C4-2B-ENZA). (**e**) Cell-doubling time was calculated for each treatment by fitting exponential growth curves using GraphPad Prism 7. Shown are means ±SEM of at least three independent experiments.

**Figure 7 cancers-12-02467-f007:**
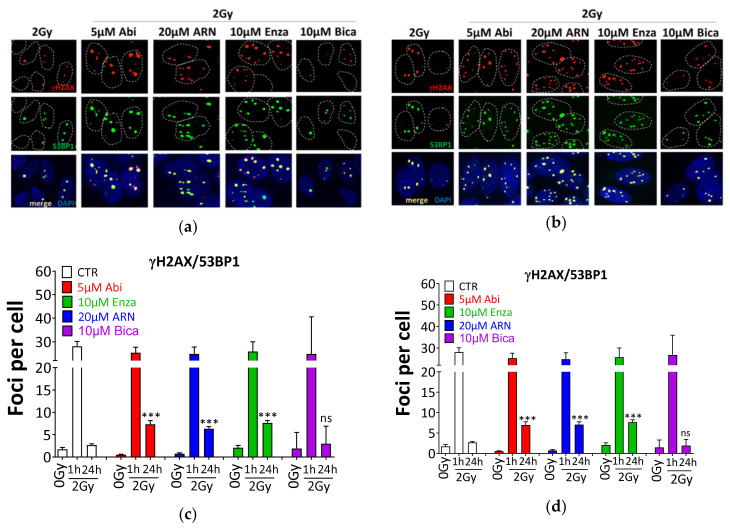
Second generation ADT inhibits the repair of ionizing radiation induced double strand breaks. Representative immunofluorescence micrographs for γH2AX (red) and 53BP1 (green) foci at 24 h after 2Gy ± the indicated antiandrogens in (**a**) LNCaP and (**b**) C4-2B cells. (**c**) and (**d**) Quantitation of colocalized γH2AX/53BP1 foci of the experiments performed in A and C, respectively. At least 100 cells were analyzed. Shown are the means ±SEM from at least three independent experiments. *p*-values were calculated using the Mann–Whitney U test. Significance is indicated as *** for *p* < 0.0001. ns: not significant.

**Figure 8 cancers-12-02467-f008:**
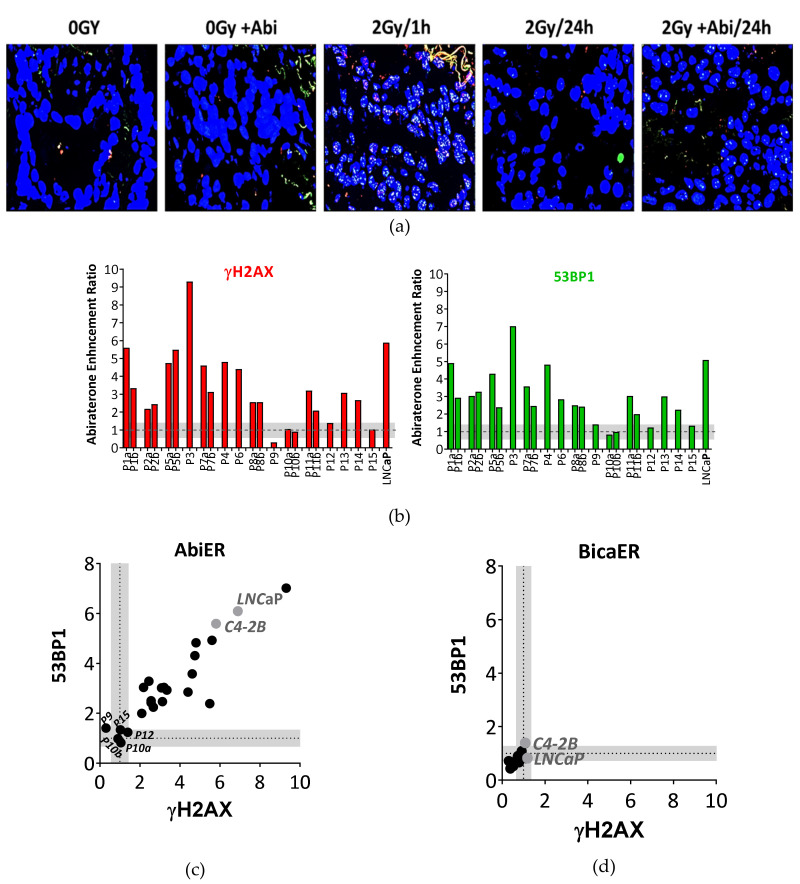
Abiraterone efficiently suppresses DSB repair in fresh PCa tissues after ex vivo irradiation. (**a**) Representative immunofluorescence micrographs for γH2AX (red) and 53BP1 (green) foci after the indicated treatments in freshly collected tumor tissue from PCa patient P5. (**b**) Abiraterone-mediated enhancement ratio (AbiER) on γH2AX (left panel) and 53BP1 (right panel) foci of 22 punch biopsies from 15 PCa patients. (**c**,**d**) Correlation between (C) AbiER or (D) BicaER on γH2AX and 53BBP1 foci. Threshold in grey was calculated as the mean standard error for each DSB marker in all samples.

**Table 1 cancers-12-02467-t001:** Patient characteristics.

Characteristics	*n*	%	Mean (±SD)/Median (Range)
Age (years)	166	100	73 (53–80)
Baseline PSA-value (ng/mL)	166	100	8 (2.1–165)
<10	100	60.2	6 (2.1–9.85)
10–20	47	28.3	12.58 (10–20)
>20	19	11.4	39 (21.5–165)
Post-therapeutic PSA-nadir (ng/mL)	164	100	0.1 (0–13.5)
Gleason-Score	166	100	7 (6–10)
<7	19	11.4	6
7	112	67,5	7
>7	35	21.1	8 (8–10)
T stage *	165 *	100	-
T1c	39	23.6	-
T2a-b	53	32.1	-
T2c-T3a/b	71	43	-
Tx	2	1.2	-
Risk categories *	166	100	-
Low risk	5	3	-
Intermediate risk	104	62.7	-
High risk	57	34.3	-
Androgen deprivation therapy	166	100	-
Yes	46	27.7	-
No	120	72.2	-
Target volume (EBRT)	166	100	-
Prostate and seminal vesicles	125	75.3	-
Additional irradiation of pelvic lymph nodes	41	24.7	-
Charlson Comorbidity Index	165	100	4 (1–9)

* For defining risk categories, classification according to NCCN was applied. Thus, the worst/highest parameter was leading in defining the underlying risk group. EBRT: External Beam Radiation Therapy; NCCN: National Comprehensive Cancer Center.

**Table 2 cancers-12-02467-t002:** Gastrointestinal toxicity (GIT).

Side Effect		ADT		Total
		No	Yes	
Acute GIT Toxicity	No GIT	48	15	63
	Grade 1 or 2 GIT	65	29	94
Total		113	44	157

**Table 3 cancers-12-02467-t003:** Genitourinary toxicity (GUT).

Side Effect		ADT		Total
		No	Yes	
Acute GUT Toxicity	No GUT	17	2	19
	Grade 1 or 2 GUT	94	42	136
Total		111	44	155

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
