# Peer review of "Second-Generation Antiandrogen Therapy Radiosensitizes Prostate Cancer Regardless of Castration State through Inhibition of DNA Double Strand Break Repair"

_cancers, 2020, doi:10.3390/cancers12092467_

Round 1
Reviewer 1 Report
The extensive inclusion of suggested further experiments, reworking of figures into the main body and appropriate citation of supporting literature, have greatly strengthened this work. I hope that the suggestion of further experiments to establish the mechanism of DNA damage response was useful - your interpretation of the data are sound and would support the observation of abiraterone contributing to an interference with HR vs NHEJ. You note in line 450-451 that this may be a greater extent of radiosensitisation from Abi - might you have any thoughts on how? This is the only question I have unanswered upon reviewing this updated manuscript.
Author Response
The extensive inclusion of suggested further experiments, reworking of figures into the main body and appropriate citation of supporting literature, have greatly strengthened this work.
Reply
The authors would like to thank the reviewer for this comment
I hope that the suggestion of further experiments to establish the mechanism of DNA damage response was useful - your interpretation of the data are sound and would support the observation of abiraterone contributing to an interference with HR vs NHEJ.
Reply
We appreciate the critical and nevertheless stimulating suggestions. After all we think that the manuscript and its message has largely benefited by these constructive comments.
You note in line 450-451 that this may be a greater extent of radiosensitisation from Abi - might you have any thoughts on how? This is the only question I have unanswered upon reviewing this updated manuscript.
Reply
A possible answer to this interesting question could be the fact the abiraterione (as you referred to in the first revision) has a dual role (AR-antagonist and CYP17 inhibitor) in suppressing the AR signaling.This explanation is now referred to in the indicated paragraph in the discussion.
Reviewer 2 Report
No further suggestions to the manuscript.
Author Response
No further suggestions to the manuscript.
Reply
Thanks for reviewing our MS
This manuscript is a resubmission of an earlier submission. The following is a list of the peer review reports and author responses from that submission.
Round 1
Reviewer 1 Report
Overall the paper is well crafted and reports some interesting data regarding the radiosensitisation and thus potential clinical utility of androgen axis agents for the treatment of prostate cancer. There are however some concerns I have regarding the use of abiraterone, the comparison of 1st gen androgen agents and the interpretation of the retrospective data.
First, abiraterone acetate is a CYP17 inhibitor, reduces production of androgen and therefore mediates control of the androgen axis and inhibits proliferation of tumours in vivo. It has been shown by others that this is not the limit of activity and in fact is a direct binder of AR and therefore can act as described in this report, as an AR antagonist. Minimally, it would warrant acknowledgment of these prior data with citation of e.g. Richards et al doi: 10.1158/0008-5472.CAN-11-3980 - beyond the one mention on line 77 of this distinct mechanism of action. Bicalutamide is mentioned in the earlier agents as are others - if this was the only drug combined with radiotherapy in the retrospective data set then its sole use as a control may be acceptable but it is not clear this is the case - if it is 6-10 times less potent, then using more should elicit a response in the cells (could this have been done), additional prostate cell lines should have been used especially as Figure 3 compares 22RV1 to LNCaP which does not seem appropriate - is the mechanism of action in cell lines the came (BIC may induce senescence or growth arrest in LNCaP (AACR report) and therefore would negate any effect of IR by preventing the cells from cycling - these data should all be generated for all cells and all drugs for a fair comparison. Figure 3, incidentally, is incorrectly labelled A, B and C whilst the text refers to A, C and D.
In fact, given the mechanism of action, might it not be considered surprising that Abi is as good, in fact better than enza or apa in enhancing the radiation induced killing - I think this is worth discussing and in context with other AR agent IR data (Goodwin is mentioned - as is the lower concentration of enza used). Though the concentrations used have been established by others, this is not covered in the discussion - are these clinically relevant/achievable?
Returning to the clinical data, if IR combinations were conducted through to 2016, were none of these performed with 2nd Gen agents? Other studies are mentioned in discussion as having favorable outcomes but the advantage in the studies here was only in biochemical recurrence, in fact ADT + IR looked to be detrimental (albeit without statistical significance) but that may have been due to elevated GUT in the combination - whilst uncomfortable to discuss in the context of potential compromise of patient benefit, this point might be worth considering in a balanced view of likely benefit of more potent inhibitors of AR and your postulated mechanism of reduced DDR - could more potent AR inhibitors worsen the adverse affects?
The link to DDR is hypothesised and explored well with damage foci experiments that are well described and analysed. A suggestion for further comparison would be to inhibit DNA-PK or HR (e.g. ATM inhibition) and see if this is 1) comparable to the effect of AR axis inhibition and 2) combines with, synergises with or detracts from the effect of e.g. IR, DNA-PK and AR antagonism. The discussion states further investigation of effect on DDR expression but that could have been included at relatively low cost with targeted qPCR post abi/enza/apa (and bic) - finding changes in expression would support the hypothesis, not doing so could suggest inhibition of DDR function. In light of the separation of hormonal signalling i.e. AR signalling using the resistance cells (again, a nice piece of work), furthering the mechanistic understanding of AR inhibition and DDR modulation is a good avenue to follow but I accept may extend beyond the scope of this work.
Specifically, in summary, I would like to see more complete bic data across cells and address the literature - describe abi function, check clinical relevance of dosing and using appropriate references - 29 and 30 are listed as supporting AR and DNA repair - androgen does not appear in either.
Author Response
Reviewer 1
Overall the paper is well crafted and reports some interesting data regarding the radiosensitisation and thus potential clinical utility of androgen axis agents for the treatment of prostate cancer.
Reply
The authors would like to thank the reviewer for this comment
There are however some concerns I have regarding the use of abiraterone, the comparison of 1st gen androgen agents and the interpretation of the retrospective data. First, abiraterone acetate is a CYP17 inhibitor, reduces production of androgen and therefore mediates control of the androgen axis and inhibits proliferation of tumours in vivo. It has been shown by others that this is not the limit of activity and in fact is a direct binder of AR and therefore can act as described in this report, as an AR antagonist. Minimally, it would warrant acknowledgment of these prior data with citation of e.g. Richards et al doi: 10.1158/0008-5472.CAN-11-3980 - beyond the one mention on line 77 of this distinct mechanism of action.
Reply
The activity of abiraterone as an AR antagonist is added to the MS as suggested with acknowledging the studies referring to this function (see P2, line 82-83).
Bicalutamide is mentioned in the earlier agents as are others - if this was the only drug combined with radiotherapy in the retrospective data set then its sole use as a control may be acceptable but it is not clear this is the case - if it is 6-10 times less potent, then using more should elicit a response in the cells (could this have been done)
Reply
In the retrospective study presented here, bicalutamide as well as gonadotropin antagonists were used as ADT. Therefore, bicalutamide was solely used in the preclinical in vitro and ex vivo studies as a control for the 1st Gen ADT, as gonadotropin antagonists are working systemically and would not work in vitro or ex vivo.
additional prostate cell lines should have been used especially as Figure 3 compares 22RV1 to LNCaP which does not seem appropriate - is the mechanism of action in cell lines the same (BIC may induce senescence or growth arrest in LNCaP (AACR report) and therefore would negate any effect of IR by preventing the cells from cycling - these data should all be generated for all cells and all drugs for a fair comparison.
Reply
We would like to thank the reviewer for this comment.
In fact, we employed 22RV1 cells not to compare them to the LNCaP cells but to investigate whether we can generalize our data to another CRPC cell lines in addition to the C4-2B cells. Further, for a direct comparison we tested our hypothesis in the hormone-resistant sub-clones raised from LNCaP and C4-2B cells.
Now the radiosensitizing effect of bicalutamide was tested in LNCaP (Fig.3C and Supp.Fig.S2C), C4-2B (Fig.3D and Supp.Fig. S2D), 22RV1 (Fig.3B) as well as in the resistant sub-clones LNCaP-Abi, LNCaP-ARN and C4-2B-ENZA (Supp.Fig.S3). The results showed a minimal synergistic effect between bicalutamide and IR only in C4-2B-ENZA cells (Supp.Fig.S3C). However, bicalutamide showed even a radio-protective effect in the other LNCaP-resistant clones LNCaP-abi (Supp.Fig.S3A) and LNCaP-ARN (Supp.Fig.S3B). This radio-protective effect can be explained by the previously described agonistic effect of bicalutamide on AR signalling, which can then lead to enhanced repair capacity and better survival after IR. The results are modified accordingly (P7, line 191-192; P8, line 230-231 and P11, line 292-302).
Moreover, the effect of bicalutamide on the repair of the IR-induced DSBs was analysed in LNCaP (Fig. 7A&C) and C4-2B (Fig. 7B&D) cells as well as in freshly collected tumor tissues from 8 different PCa patients (Fig. 8 and Supp. Fig. S8). Our results further confirmed the absence of a radiosensitizing effect of bicalutamide as evidenced by no significant change in the number of residual gH2AX/53BP1 in all systems analysed.
Figure 3, incidentally, is incorrectly labelled A, B and C whilst the text refers to A, C and D.
Reply
Now Fig. 3 is correctly labelled and referred to in the MS.
In fact, given the mechanism of action, might it not be considered surprising that Abi is as good, in fact better than enza or apa in enhancing the radiation induced killing - I think this is worth discussing and in context with other AR agent IR data (Goodwin is mentioned - as is the lower concentration of enza used). Though the concentrations used have been established by others, this is not covered in the discussion - are these clinically relevant/achievable?
Reply
We agree with the reviewer that Abi can be as good or even better than ENZA and APA in potentiating the IR cytotoxicity. In fact, this is the top message of the current study. We showed that combining either of the aforementioned 2nd Gen ADTs has efficiently radiosenitized different PCa cells, regardless of the castration state.
The concentrations of the ADT used for present pre-clinical in vitro and ex vivo study are in the range of clinically achievable steady state plasma concentrations (Cmax) (doi:10.1158/1078-0432.CCR-16-3083). Consequently, this study provides the evidence base for the clinical use of the aforementioned drugs as radiosensitizers in prostate cancer. This has now been included in the revised manuscript.
Returning to the clinical data, if IR combinations were conducted through to 2016, were none of these performed with 2nd Gen agents? Other studies are mentioned in discussion as having favorable outcomes but the advantage in the studies here was only in biochemical recurrence, in fact ADT + IR looked to be detrimental (albeit without statistical significance) but that may have been due to elevated GUT in the combination - whilst uncomfortable to discuss in the context of potential compromise of patient benefit, this point might be worth considering in a balanced view of likely benefit of more potent inhibitors of AR and your postulated mechanism of reduced DDR - could more potent AR inhibitors worsen the adverse effects?
Reply
In our retrospective analysis no 2nd GEN agent was used, only bicalutamide or gonadotropin antagonists.
The reviewer raised an interesting point concerning the side effects of the used drugs, given that they inhibit the DDR. Therefore, we think that a phase I study should be initiated. However, given that the IR is a local therapy with side effects restricted to tissues lacking AR and that the 2nd gen ADT are already tested in vivo and approved by the FDA as well as the concentrations used here are within the range of the Cmax, we do not think that the side effects would be greatly affected. This has now been included in the revised MS.
The link to DDR is hypothesised and explored well with damage foci experiments that are well described and analysed. A suggestion for further comparison would be to inhibit DNA-PK or HR (e.g. ATM inhibition) and see if this is 1) comparable to the effect of AR axis inhibition and 2) combines with, synergises with or detracts from the effect of e.g. IR, DNA-PK and AR antagonism. The discussion states further investigation of effect on DDR expression but that could have been included at relatively low cost with targeted qPCR post abi/enza/apa (and bic) - finding changes in expression would support the hypothesis, not doing so could suggest inhibition of DDR function. In light of the separation of hormonal signalling i.e. AR signalling using the resistance cells (again, a nice piece of work), furthering the mechanistic understanding of AR inhibition and DDR modulation is a good avenue to follow but I accept may extend beyond the scope of this work.
Reply
We are in fact currently addressing this question in more detail in a separate project. According to the preliminary data of this project, we do believe the targeted DSB repair pathway - by at least abiraterone - is the homologous recombination repair pathway. However, for more convenience we now added the requested experiment by combining either ATM or DNA-PK inhibitor together with abiraterone to investigate which inhibitor results in a synergistic effect with abiraterone. As illustrated in Supp.Fig.S5, combining ATMi and Abi did not further increase the number of residual gH2AX/53BP1 compared to either ATMi or Abi alone. This indicates that the inhibited pathway is probably the ATM-dependent HR pathway. In agreement with this assumption, DNAPK inhibition increased the number of residual gH2AX/53BP1 significantly more than the increase mediated by Abi alone. Interestingly, combining DNAPKi and Abi showed a tendency of a synergistic increase in the number of residual unrepaired DSBs.
Specifically, in summary, I would like to see more complete bic data across cells and address the literature - describe abi function, check clinical relevance of dosing and using appropriate references - 29 and 30 are listed as supporting AR and DNA repair - androgen does not appear in either.
Reply
Collectively, we deeply thank the reviewer for these constructive comments and hope we addressed all their concerns.
Now the references are correctly cited.
Reviewer 2 Report
This is an excellent manuscript of high clinical significance as RT is a curative treatment of choice vs. surgery in localized PCa. The goal of the study is to evaluate novel anti-androgens for combination treatment with RT. The use of second generation is increasing in localized PCa. The design of the study is sound and interesting as the comparison between the first and second generation antiandrogens is included. This is preclinical study involving several models of PCa cell lines that represent heterogenity of PCa and hormone dependence.
One downside of the study is lack of experiments involving mouse models, specifically xenografts. The authors should discuss these types of studies done with RT/ADT - are they useful or not? In thar context the ex vivo studies are interesting and significant.
Readability and impact of the paper could be improved by more clear labeling of graphs (larger font). This applies to most of figures and graphs.
Author Response
Reviewer 2
This is an excellent manuscript of high clinical significance as RT is a curative treatment of choice vs. surgery in localized PCa. The goal of the study is to evaluate novel anti-androgens for combination treatment with RT. The use of second generation is increasing in localized PCa. The design of the study is sound and interesting as the comparison between the first and second generation antiandrogens is included. This is preclinical study involving several models of PCa cell lines that represent heterogenity of PCa and hormone dependence.
Reply
The authors would like to thank the reviewer for this comment
One downside of the study is lack of experiments involving mouse models, specifically xenografts. The authors should discuss these types of studies done with RT/ADT - are they useful or not? In thar context the ex vivo studies are interesting and significant.
Reply
We are currently addressing this question in a parallel project.
Readability and impact of the paper could be improved by more clear labeling of graphs (larger font). This applies to most of figures and graphs.
Reply
Now better and clearer graphs (with bigger fonts) are uploaded.